# Symptom burden according to dialysis day of the week in three times a week haemodialysis patients

**Pann Ei Hnynn Si**[1]*, **Rachel Gair**[2], **Tania Barnes**[1], **Louese Dunn**[1], **Sonia Lee**[1], **Steven Ariss**[3], **Stephen J. Walters**[3], **Martin Wilkie**[1], **James Fotheringham**[1,3]

**1** Sheffield Kidney Institute, Sheffield Teaching Hospitals NHS Foundation Trust, Sheffield, United Kingdom, **2** Think Kidneys, UK Renal Registry, Bristol, United Kingdom, **3** School of Health and Related Research, University of Sheffield, Sheffield, United Kingdom

* pann-ei.hnynn-si@nhs.net

**Data Availability Statement:** A minimal dataset required to reach the conclusions drawn from this manuscript required the linkage of identifiable patient information collected during the trial to

## Abstract

### Background

Haemodialysis patients experience significant symptom burden and effects on health-related quality of life. Studies have shown increases in fluid overload, hospitalization and mortality immediately after the long interdialytic interval in thrice weekly in-centre haemodialysis patients, however the relationship between the dialytic interval and patient reported outcome measures (PROMs) has not been quantified and the extent to which dialysis day of PROM completion needs to be standardised is unknown.

### Methods

Three times a week haemodialysis patients participating in a stepped wedge trial to increase patient participation in haemodialysis tasks completed PROMs (POS-S Renal symptom score and EQ-5D-5L) at recruitment, six, 12 and 18 months. Time from the long interdialytic interval, HD day of the week, and HD days vs non-HD days were included in mixed effects Linear Regression, estimating severity (none to overwhelming treated as 0 to 4) of 17 symptoms and EQ-5D-5L, adjusting for age, sex, time on HD, control versus intervention and Charlson Comorbidity Score.

### Results

517 patients completed 1659 YHS questionnaires that could be assigned HD day (510 on Mon/Tue/Sun, 549 on Wed/Thu/Tue, 308 on Fri/Sat/Thu and 269 on non-HD days). With the exception of restless legs and skin changes, there was no statistically significant change in symptom severity or EQ-5D-5L with increasing time from the long interdialytic interval. Patients who responded on non-HD days had higher severity of poor appetite, constipation, difficulty sleeping, poor mobility and depression (approximately 0.2 severity level), and lower EQ-5D-5L (-0.06, CI -0.09 to -0.03) compared to HD days.

Hospital Episode Statistics data, which at the time of writing is provided by the NHS Digital Data Access Request Service (NHS DARS, https://digital.nhs.uk/services/data-access-request-service-dars), and then appropriate processing. An application to NHS DARS can be submitted detailing lawful processing of the combined dataset and the period which HES data is required for. NHS DARS would verify appropriate permissions were in place as a result of this process. A data sharing agreement between the relevant parties would allow data to be transferred from the University of Sheffield to NHS DARS and on to those wishing to perform the enclosed analyses. Please contact ctru@sheffield.ac.uk for further information about the unlinked dataset which has the personal information required for linkage.

**Funding:** The Health Foundation (Scaling Up Round 2) funded the SHAREHD study and had no role in its design, data collection, analysis, interpretation, decision to publish or preparation of the manuscript.

**Competing interests:** PH conducts research funded by Vifor Pharma. JF has received speaker honoraria from Fresenius medical care, and conducts research funded by the National Institute of Health Research (NIHR), Vifor Pharma and Novartis. MEW has received speaker honoraria Fresenius and Baxter, has acted on an advisory board for Baxter and has conducted research funded by NIHR. SJW has received book royalties from Wiley and has received funds from NIHR, the Department of Health and Medical Research Council. This does not alter our adherence to PLOS ONE policies on sharing data and materials.

## Conclusions

Measuring symptom severity and EQ-5D-5L in haemodialysis populations does not need to account for dialysis schedule, but completion either on HD or non-HD days could introduce bias that may impact evaluation of interventions. Researchers should ensure completion of these instruments are standardized on either dialysis or non-dialysis days.

## Introduction

People receiving haemodialysis (HD) for end stage renal failure (ESRF) experience significantly impaired health-related quality of life (HRQoL) and high symptom burden [1]. When looking to improve outcomes on haemodialysis, patients, clinicians and policy-makers are increasingly focusing on HRQoL and symptoms in addition to the traditional endpoints including cardiovascular disease and survival [2]. Patients state HRQoL is a basic aspect of health [3], and studies have shown these patient-reported outcomes measures (PROMs) are more strongly associated with the risk of death and hospitalization than clinical parameters [4]. Although use of PROMs in haemodialysis studies are increasing, standardisation in terms of how measures are implemented is lacking. In addition to the unmet need is the undetected need: healthcare providers fail to commonly recognise and treat the physical and emotional symptoms experienced by HD patients [5, 6]. Significant changes in the severity of symptoms occur at a median of 3 months, justifying regular surveillance of symptoms in the dialysis population [7], interventions based on monitoring symptoms to improve outcomes in haemodialysis patients [8], and the robust measurement of symptoms in these settings.

In addition to the response of symptom severity to interventions in the clinical or experimental setting over time, the HD schedule may introduce further variation. It is recognised that the varying intervals between haemodialysis sessions have an association with volume status, uraemia and electrolyte imbalance, measures of cardiac function, hospitalisation and death [9, 10]. When Standardized Outcome in Nephrology (SONG-HD), a consensus exercise to establish core outcomes to be measured and reported in haemodialysis trials, spoke to patients about their experience of fatigue, they reported "extreme fatigue on Sunday night because it was 2.5 days without treatment" [2]. Furthermore patients' perceptions of their symptoms or quality of life may be more important than objective clinical assessments using validated instruments [11]. There is expanding literature that PROMs are not only effected by psychosocial issues, stress, emotions, patient characteristics and co-morbidities [12, 13], the environment in which the instrument is completed may influence the result [11].

HRQoL has been shown to be a predictor of morbidity and mortality in haemodialysis patients [14, 15] and HRQoL measures play an important role in evaluating cost effectiveness of treatment. Failure to account for any underlying differences in symptom severity due to the day of instrument completion in relation to the dialysis schedule could bias the impact and effectiveness of interventions for symptoms and HRQoL, which have been prioritised by the patients and clinicians [2] and could lead to failure of new treatment or interventions to be approved. To quantify this, we used data from a large stepped wedge randomised controlled trial with aim to determine the association between symptom burden and the haemodialysis schedule in three times a week haemodialysis patients and to explore the effect of PROMs completion on dialysis and non-dialysis days, accounting for patient characteristics.

## Material and methods

### Study design

Data for this study was obtained from SHAREHD Stepped Wedge Cluster Randomised Trial (SWCRT) [16, 17]. which evaluated a quality improvement collaborative designed to create an environment to support in-center HD patients to dialyse more independently. The evaluation ran for 18 months with an additional six months sustainability, and was conducted in 12 renal centres in England. It ran from October 2016 to October 2018: following a control period of six months (baseline), six centres participated in the intervention immediately (step one) with six centres joining after a further six months (step two).

### Setting

Trained, delegated research nurses gained written informed consent to participate from prevalent HD patients established on centre-based haemodialysis. The study adhered to the declaration of Helsinki, ethical approval was obtained from West London & GTAC Research Ethics Committee (IRAS project ID 212395) and the trial was registered (ISRCTN Number 93999549). This presented analysis was specified in the research protocol [16].

  The recruitment exclusion criteria were patients too unwell to engage in the study, as judged by the clinical team, or patients unable to understand written and verbal communication in English. For this specific analysis, we excluded patients who had missing data for the adjustment covariates, did not dialyse three times a week, or we could not assign a HD schedule for (Fig 1). Failure to link comorbidity using National Health Service (NHS) number by NHS digital resulted in missing comorbidity data and was assumed at random. Centres participating in the trial were: Sheffield Teaching Hospital NHS Foundation Trust, Central Manchester Healthcare Trust, City Hospitals Sunderland NHS Foundation Trust, East & North Hertfordshire NHS Trust, Guy's & St Thomas NHS Foundation Trust, Heart of England Foundation Trust, Leeds teaching Hospitals NHS Trust, The Royal Wolverhampton NHS Trust, North Bristol NHS Trust, University Hospital of North Midlands NHS Trust, Nottingham University Hospitals NHS Trust.

### Participants and data collection

Patients participating in SHAREHD were asked to complete instruments at baseline, six, 12, and 18 months at either dialysis unit or home or clinic. They were allowed to complete these instruments freely at the time and day they choose. A delegated member of the local research team collected research nurse-completed and self-completed paper instruments, which included patient demography and dialysis schedules. The Think Kidneys Your Health Survey (YHS) includes the POS-S Renal [18] and EQ-5D-5L [19]: POS-S renal consists of 17 symptoms commonly experienced by HD patients: weakness, poor mobility, pain, difficulty sleeping, breathlessness, drowsiness, feeling anxious, itching, dry mouth, restless legs, feeling depressed, poor appetite, changes in skin, constipation, nausea, diarrhoea and vomiting. Each symptom is scored on a 5-level ordinal scale ranging from not at all to overwhelmingly. The EQ-5D-5L is a five-item standardized instrument developed as a measure of generic health-related quality of life and it consists of a descriptive system which comprises five dimensions: mobility, self-care, usual activities, pain/discomfort and anxiety/depression, for which respondents select one of five levels from no problems to unbearable or being unable to perform the domain. From these responses a utility scale can be calculated from zero to one: zero being death and one being in perfect health. This generic measure and associated utility value are used to compare health between groups of individuals with chronic conditions, particularly for health economic evaluations.

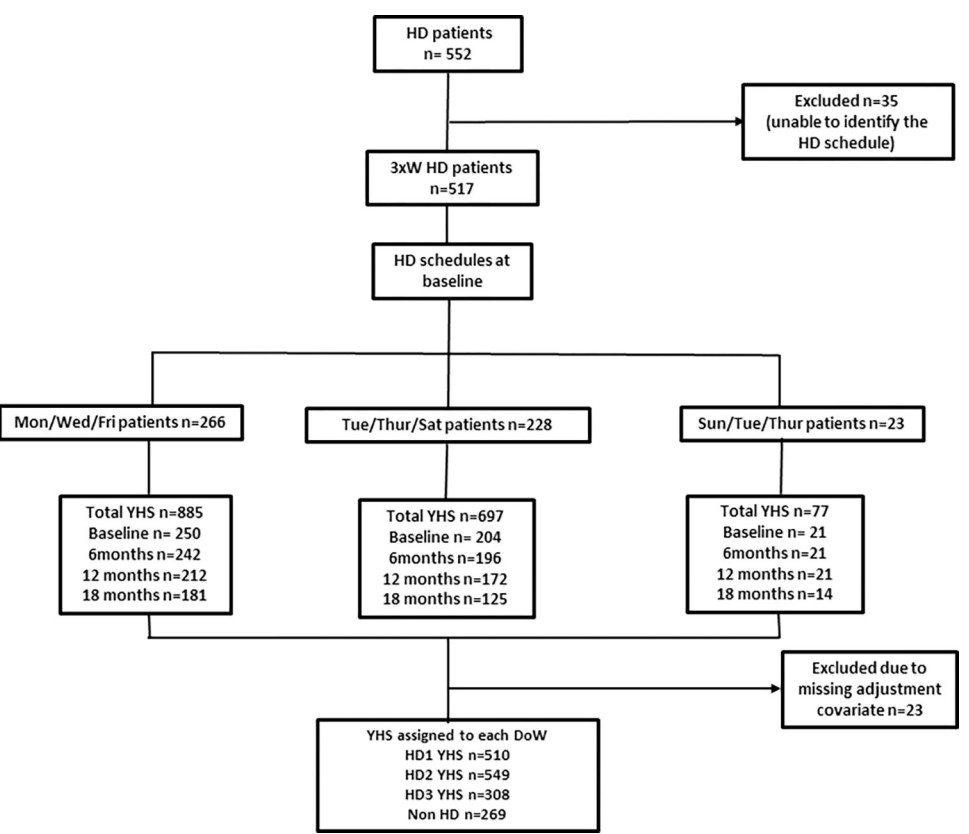

**Fig 1. Flow diagram of the participants and completed questionnaires in the study.**

Sociodemographic variables including age, gender, ethnicity and education were also collected. The Modified End Stage Renal Disease (ESRD) Charlson comorbidity index (CCI) score [14, 20] was calculated using established algorithms and weights from hospitalisation data obtained through linkage to hospital episode statistics by the NHS Digital Data Access Request Service.

## Study size

The informing SWCRT sample size was determined using a recommended ICC value of 0.05 [21], to have a 90% power to detect an increase in the proportion undertaking five or more haemodialysis-related tasks from 15% to 30% with statistically significance at the 5% two-sided level: 12 clusters of 25 patients, with six clusters randomised at each step of SWCRT was arrived at. In recognition of the background mortality and renal transplantation rate and to mitigate the risk of incomplete data collection, the target recruitment per participating site was doubled to 50 [17].

Statistical analysis

Dialysis day of the week (the day of the week in relation to HD schedule) was determined from the date that the instrument was completed, combined with the dialysis schedule (routine attendance on Mon/Wed/Fri, Tue/Thu/Sat or Sun/Tue/Thu) reported at that study timepoint. Descriptive statistics were used to summarize the frequency and distribution of the questionnaires and other baseline characteristics in the study cohort, with symptoms reported recoded as none vs mild to overwhelming based on the distribution of responses. As visually a number

of symptoms showed a progressive improvement from the long interdialytic interval, in the primary analysis, dialysis day of the week was treated as a continuous variable. The change in symptom severity was assessed for each additional dialysis session after the two-day break: Mon/Wed/Fri HD patients completing their questionnaires on Monday were assigned to HD day 1 (HD1), with Wednesday responses to HD day 2 (HD2) and Friday responses to HD day 3 (HD3) and therefore, HD1 was assigned to those who dialyzed and completed the instruments on Mon/Tue/Sun, HD2 to Wed/Thu/Tue and HD3 to Fri/Sat/Thu respectively. Patients who completed instruments on non-dialysis days were assigned to non-HD day.

Mixed effects linear regression was employed, with the patient specified as a random intercept, therefore allowing changes in symptom burden according to the dialysis schedule to be estimated within an individual. For the purposes of statistical analysis, the five-point responses to the symptom questions were coded 0 = not at all; 1 = slightly; 2 = moderately; 3 = severely and 4 = overwhelmingly. With the symptom recoded 0 to 4, the linear mixed effects model reports a coefficient representing the size of the change in severity level associated with a variable. e.g., a value of 1.0 would represent a change from none to mild or severe to overwhelming. Models predicting severity of each symptom were adjusted for age (<40, 40–65, >65), sex, years receiving dialysis (less than 1 year, 1–5 years, more than 5 years), comorbidity (Modified ESRD Charlson Index Score 0, 1–5 or more than 5) and steps (control arm versus intervention arm). Years on dialysis was defined as the time between the first day of dialysis treatment and the study entry date.

Sensitivity analyses exploring the relationship between symptoms and dialysis schedule were performed: HD2 and HD3 were compared individually to HD1 as a categorical variable, and HD days were compared to non-HD days. Additional analyses exploring the symptom burden affected by patient baseline characteristics and co morbidities was performed adjusted for dialysis day of the week treating HD days as continuous variable. P value < 0.05 was considered the threshold for statistical significance. All analyses were carried out using SPSS 26.

## Results

Of the 586 haemodialysis patients recruited to the SHAREHD trial, 552 in-centre HD patients from the 12 participating renal centres provided data during the baseline phase. Thirty-five participants who were not dialysing 3 times a week (3xW) were excluded (6.3%). 1659 YHS questionnaire instruments were completed by 517 3xW participants at baseline, 6 and 12 months, 18 months. HD schedule at the beginning of the study was Mon/Wed/Fri (n = 266, 48.2%), Tue/Thu/Sat (n = 228, 41.3%) and Sun/Tue/Thu (n = 23, 4.3%) (Fig 1). 32 patients (6%) changed their HD schedule during the study. There were 475 observations at baseline control period (step 0), 459 observations at step 1 (212 observations in control group and 247 in intervention group) and 725 observations at step 2 (intervention group).

Out of the YHS questionnaires completed by 517 participants, 510 were assigned to HD1 (Mon/Tue/Sun), 549 to HD2 (Wed/Thu/Tue), 308 to HD3 (Fri/Sat/Thu) and 269 to non-HD day. 23 YHS questionnaires were excluded due to missing adjustment covariates.

### Patient's characteristics

The demography of patients stratified with haemodialysis schedules are outlined in Table 1 and patients who dialysed on Mon/Wed/Fri and Tue/Thu/Sat are younger but have higher comorbidities scores compared to patients who dialysed on Sun/Tue/Thu. The majority of patients were male (61%) with a mean age of 63 years, with 23.2% having been on haemodialysis for less than a year. Caucasian patients accounted for 81.2% of the patients. The Mean Modified Charlson score index was 2.85 (SD 2.77).

**Table 1. Demography of patients stratified by haemodialysis pattern at baseline.**

| | | Mon/Wed/Fri | Tue/Thu/Sat | Sun/Tue/Thu | Total |
|---|---|---|---|---|---|
| **Number of haemodialysis patients** | | N = 266 | N = 288 | N = 23 | N = 517 |
| **Total number of observations** | | 885 | 697 | 77 | 1659 |
| **Mean Age** | | 61.5 (SD 15.51) | 64.6 (SD 15.40) | 66 (SD 16.29) | 63 (SD 15.56) |
| **Sex** | Male | 61% (158/259) | 62% (134/216) | 57.1% (12/21) | 61.3% (304/496) |
| **Ethnicity** | White | 81.7% (210/257) | 80.7% (171/212) | 81% (17/21) | 81.2% (398/490) |
| **Education** | No formal education | 29.2% (74/253) | 38.5% (82/213) | 57.1% (12/21) | 34.5% (168/487) |
| | High education (1–3)* | 49.4% (125/253) | 42.3% (90/213) | 23.8% (5/21) | 45.2% (220/487) |
| | Higher education (4–6)** | 21.3% (54/253) | 19.2% (41/213) | 19.0% (4/21) | 20.3% (99/487) |
| **Myocardial infarction** | | 17.2% (42) | 23.1% (48) | 9.5% (2) | 19.5% (92) |
| **Heart Failure** | | 19.3% (47) | 18.8% (39) | 23.8% (5) | 19.2% (91) |
| **Cancer** | | 7.8% (19) | 9.1% (19) | 4.8% (1) | 8.2% (39) |
| **Connective Tissue Disease** | | 7.4% (18) | 2.4% (5) | 4.8% (1) | 5.1% (24) |
| **Cerebral vascular accident** | | 7.4% (18) | 9.6% (20) | 0.0% (0) | 8% (38) |
| **DM without complication** | | 37.3% (91) | 37.0% (77) | 33.3% (7) | 37% (175) |
| **DM with complication** | | 23.8% (58) | 26.0% (54) | 9.5% (2) | 24.1.% (114) |
| **Pulmonary Disease** | | 20.1% (49) | 23.6% (49) | 9.5% (2) | 21.2% (100) |
| **Peripheral vascular disease** | | 27.0% (66) | 25.0% (52) | 14.3% (3) | 25.5% (121) |
| **Severe Liver disease** | | 0.8% (2) | 1.0% (2) | 0.0% (0) | 0.8% (4) |
| **Lymphoproliferative disease** | | 1.2% (3) | 1.0% (2) | 0.0% (0) | 1.1% (5) |
| **Metastatic cancer** | | 0.8% (2) | 0.8% (2) | 0.0% (0) | 0.8% (4) |
| **Paraplegia** | | 0.8% (2) | 2.4% (5) | 0.0% (0) | 1.5% (7) |
| **Modified Charlson score index (score 0–16)*** | Score 0 | 23.8% (58) | 22.1% (46) | 38.1% (8) | 23.7% (112) |
| | Score 1–5 | 63.1% (154) | 60.6% (126) | 61.9% (13) | 61.9% (293) |
| | Score >5 | 13.1% (32) | 17.3% (36) | 0% (0) | 14.4% (68) |
| **Years on dialysis** | <1yr on RRT | 17.9% (40) | 28.3% (51) | 36.8% (7) | 23.2% (98) |
| | 1–5 year | 52% (116) | 43.3%(78) | 57.9% (11) | 48.6% (205) |
| | >5 years | 30% (67) | 28.3% (51) | 5.3% (1) | 28.2% (119) |

*High education (1 = professional qualification, 2 = 'O' level/GSCE equivalent,3 = Apprenticeship).

**Higher education (4 = 'A'level/higher equivalent,5 = Degree or higher, 6 = Diploma).

*** Higher Modified Charlson score indicates high comorbidities.

## Prevalence of symptoms and association with patient characteristics

Table 2 presents symptom prevalence at baseline defined as the proportion of patients with presence of symptoms (mild, moderate, severe or overwhelming symptoms), stratified by hae-modialysis schedules that patients are receiving. Fig 2 shows the severity of each of the 17 symptoms, stratified by HD Days (Dialysis Day of the week: HD1, HD2, HD3) (S1 Table).

In descending frequency, the five most prevalent symptoms were: weakness (82.5%), poor mobility (69.3%), drowsiness (67.1%), difficulty in sleeping (66.1%) and itching (65.5%) (Table 2). Although symptom prevalence varies by HD schedule patients were receiving, patients who dialysed on Sun/Tue/Thu have less symptom prevalence in 11 out of 17 symptoms compared to other two HD schedules (Table 2). Higher symptom burden was found only for breathlessness, poor appetite, itching, difficulty sleeping, diarrhoea and restless legs in Sun/Tue/Thu HD schedule. The multivariable adjusted association between patient characteristics and each of the symptoms in the POS-S renal and additionally adjusted for dialysis day of the week (HD1, HD2, HD3) is reported separately (S2 Table). Age, sex, years on dialysis and comorbidities independently predict symptom severity. With the exception of skin changes

**Table 2. Symptoms prevalence at baseline (mild or worse) stratified according to haemodialysis schedule.**

|  | Mon/Wed/Fri | Tue/Thu/Sat | Sun/Tue/Thu | Total |
|---|---|---|---|---|
| **Number of Patients** | n = 266 | n = 228 | n = 23 | n = 517 |
| **eq5d5l** | 0.68 | 0.69 | 0.74 | 0.70 |
| **Pain** | (160) 62.0% | (138) 63.0% | (14) 60.9% | (312) 62.4% |
| **Breathlessness** | (146) 57.3% | (126) 57.0% | (14) 60.9% | (286) 57.3% |
| **Weakness** | (219) 84.9% | (179) 80.3% | (18) 78.3% | (416) 82.5% |
| **Nausea** | (99) 38.2% | (87) 39.5% | (9) 39.1% | (195) 38.8% |
| **Vomiting** | (54) 20.8% | (52) 23.5% | (4) 17.4% | (110) 21.9% |
| **Poor appetite** | (129) 50.0% | (110) 49.5% | (12) 52.2% | (251) 49.9% |
| **Constipation** | (97) 37.5% | (80) 36.5% | (6) 27.3% | (183) 36.6% |
| **Sore mouth** | (119) 45.8% | (117) 52.9% | (9) 39.1% | (245) 48.6% |
| **Drowsiness** | (165) 64.2% | (157) 70.7% | (15) 65.2% | (337) 67.1% |
| **Poor mobility** | (177) 68.6% | (159) 71.0% | (14) 60.9% | (350) 69.3% |
| **Itching** | (169) 65.0% | (142) 64.8% | (18) 78.3% | (329) 65.5% |
| **Difficulty sleeping** | (170) 65.6% | (146) 65.5% | (18) 78.3% | (334) 66.1% |
| **Restless legs** | (138) 53.5% | (114) 51.8% | (16) 69.6% | (268) 53.5% |
| **Changes in skin** | (132) 51.4% | (106) 48.2% | (11) 50.0% | (249) 49.9% |
| **Diarrhoea** | (82) 31.7% | (53) 24.2% | (9) 39.1% | (144) 28.7% |
| **Anxious** | (132) 51.0% | (105) 47.3% | (10) 43.5% | (247) 49.0% |
| **Depressed** | (125) 48.4% | (108) 48.6% | (9) 39.1% | (242) 48.1% |

which were significantly higher in female, there were no significant differences in other symptom burden between males and females (S2 Table). Compared to age group 40–65 years old, older patients (>65 years) have statistically significant less symptoms burden in a range of symptoms: pain, weakness, nausea, vomiting, poor appetite, constipation, drowsiness, difficulty sleeping, restless legs, anxiety, and depression. Symptom severity was higher in a range of symptoms in respondents who had received dialysis for more than five years and had greater comorbidity defined by a Charlson comorbidity score of 5 or more. Importantly, no patient characteristic resulted in an increase in symptom severity that exceeded 1.0, representing a change in level of severity (e.g., from moderate to severe). There were 62 participants who completed only one instrument throughout the study period. Comparing this cohort with patients who completed the instruments more than once in follow up period, patient characteristics and severity of symptom burden at baseline were similar (S3 and S4 Tables).

## Effect of symptom burden and change in symptom score according to dialysis day of the week

Fig 3 and S5 Table report the changes in symptom severity associated with increasing time from the long interdialytic interval, HD days compared to non-HD days, and individual HD days (HD1 vs HD2 and HD1 vs HD3). Estimating symptom changes over time after 2-day break (time from HD1) using multivariable mixed effects linear regression analysis adjusted for age, sex, comorbidity, control versus intervention period, and time on dialysis revealed that among 17 symptoms of POS-S renal, only restless legs (effect size 0.1, P = 0.014, 95% CI 0.02 to 0.18) and changes in skin (effect size 0.08, P = 0.03, 95% CI 0.01 to 0.16) significantly worsened with increasing time from 2-day break (S5 Table). Increasing time from the long interdialytic interval (e.g., 0, 2 and 4 days) was not significantly associated with a change in severity for remaining 15 symptoms. Control compared to intervention period did not significantly influence the severity of any of the 15 symptoms.

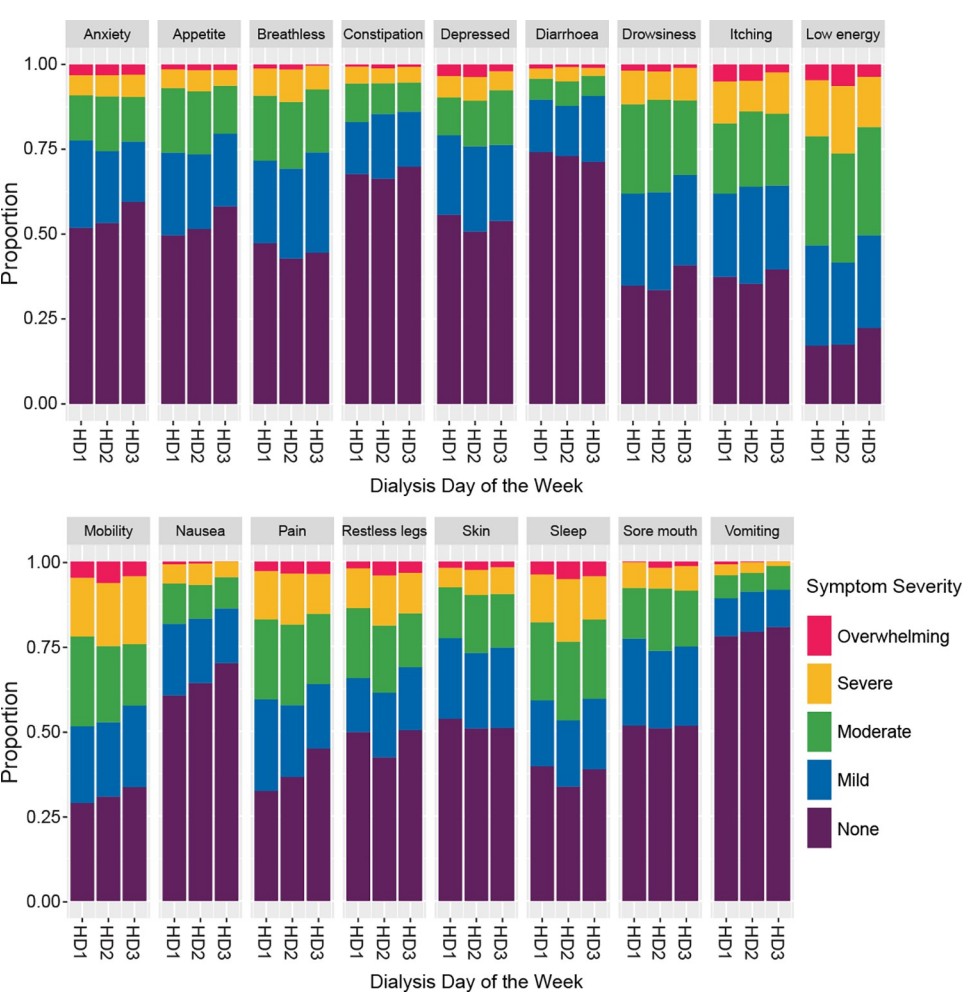

**Fig 2. Symptom severity stratified by HD day (dialysis day of the week).** The observations from 4 time points (baseline, six, 12 and 18 months) were used to inform this figure (S1 Table).

Compared to HD days, the severity of responses on non-HD days were significantly higher for poor appetite (effect size 0.2, P = 0.004, 95% CI 0.06 to 0.34), constipation (effect size 0.19, P = 0.005, 95% CI 0.06 to 0.31), difficulty sleeping (effect size 0.2, P = 0.012, 95% CI 0.04 to 0.36) and depression (effect size 0.17, P = 0.016, 95% CI 0.03 to 0.31) (Fig 3) (S5 Table).

Comparing the HD days individuals to each other, patient's symptoms worsen significantly on the second HD day (HD2) compared to the first (HD1) for breathlessness (effect size 0.17, P = 0.004, 95% CI 0.05 to 0.28), weakness (effect size 0.2, P = 0.002, 95% CI 0.07 to 0.33), difficulty sleeping (effect size 0.23, P = 0.001, 95% CI 0.09 to 0.36), restless legs (effect size 0.18, P = 0.006, 95% CI 0.05 to 0.32) and changes in skin (effect size 0.13, P = 0.04, 95% CI 0.01 to 0.26) (Fig 3) (S5 Table). Comparing HD1 and HD3, we found that restless legs and changes in skin worsen significantly on HD3. None of the symptoms improved significantly in both HD2 and HD3 comparing categorically to HD1.

Overall, the mean EQ-5D-5L score was 0.70 (1 being perfectly healthy and 0 being dead) (Table 2). There was no significant change in EQ-5D-5L score with increasing time from the long interdialytic interval and comparing HD2 and HD3 to HD1 (Fig 3) (S5 Table). However, EQ-5D-5L was significantly lower in non-HD days compared to HD Days (effect size -0.06, 95% CI -0.09 to -0.03) (Fig 3) (S5 Table).

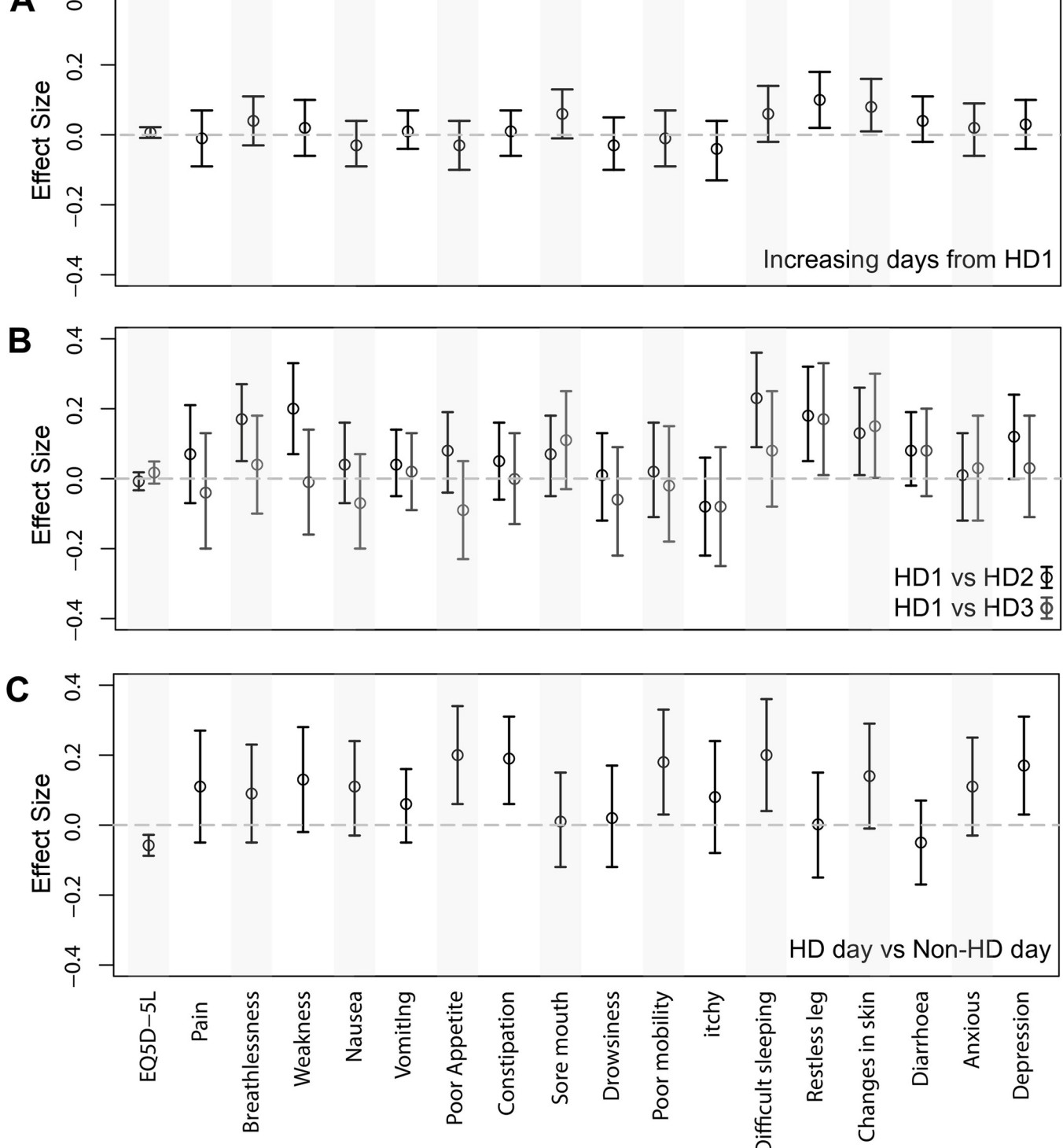

**Fig 3. Multivariable mixed effects linear regression comparing each hemodialysis day after long break, each HD day from HD1, HD day vs non-HD day (S5 Table).**

## Discussion

This observational study evaluated the symptom burden associated with dialysis day of the week in three times a week haemodialysis patients using POS-S Renal and EQ-5D-5L [19] at baseline, six months, 12 months and 18 months. The baseline characteristics are comparable to national registry data describing prevalent patients [22] and high levels of comorbidity were reported. This analysis showed that with the exception of restless legs and changes in skin, symptom severity was not significantly different with increasing time from two-day break although milder symptom severity on the first HD day compared to other HD days was observed in breathlessness, weakness, difficulty sleeping, restless legs and changes in skin. Symptom burden was significantly higher and EQ-5D-5L was significantly lower in non-HD days compared to HD days. No patient characteristic resulted in an increase in symptom severity that exceeded 1.0, representing a change in level of severity (e.g., from moderate to severe) although older patients (>65yrs) reported less symptoms and respondents receiving HD more than five years and more comorbid respondents reported more symptoms which reached statistical significance.

Symptoms prevalence in this study is comparable to other studies [23] including a systematic review on symptoms prevalence in ESRF patients [24], for prevalence, type of symptoms reported and cohort mean HRQoL. Variation in severity of symptoms in relation to patient characteristics and comorbidities was also comparable [13]. Despite associations between the interdialytic interval and a range of outcomes, although we observed trends, we did not find statistically significant variation in most symptoms over the dialysis day of the week. Explanations for this include changes in volume, cardiac and uraemic markers being of insufficient size to manifest in PROMs, and inadequate sample size. Change in effect size for symptoms were less than 0.5 points on the scale of 0 (none) to 4 (overwhelming): Using distribution-based method ½ SD [25], the observed changes are less than minimum importance difference (MID) although they statistically significant. Although EQ5D-5L was lower in non-HD days, it is less than mean minimum important difference (MID) estimate of 0.074 (range -0.011 to 0.140) for the three-level version of the EQ-5D from eight non-renal longitudinal studies in eleven patient groups [26].

The strengths of this study are that it tackles a previously unexplored question, which could have altered practice in the range of ongoing clinical trials aiming to improve symptoms, and that this study repeatedly sampled individuals over an 18 months period for a range of symptoms. Weaknesses include the assumption of linearity across the levels of severity, which we used visual inspection of the underlying trends (Fig 2) and the application of exploratory non-linear methods during the development of the study to justify. For instruments completed on a dialysis day, we were unable to determine the timing of the completion of the instruments in relation to the dialysis session which can be either before or during or after dialysis session. This may be relevant as patients may feel more unwell immediately after dialysis due to dialysis process itself affecting their symptom burden. However, it is our experience that most patients completed the instruments once they established on the haemodialysis machine. As the participants were allowed to complete the instruments on the day of their choice either on non-dialysis day or dialysis day and they may choose to complete these questionnaires on the day they feel better, this has potential impact on the outcome of symptom burden in relation to dialysis days. We performed multiple testing with different variables with statistical significance considered as a p value less than 0.05, and concluded adjustment for dialysis day of the week is not required as most of the results were non-significant. Correction of p values for multiple testing would further reduce significance, not altering our conclusions, and is therefore is not required.

We identified significant associations between symptom severity in different age groups, comorbidity and time on dialysis, mandating their adjustment in observational studies and potentially in clinical trials as studies have shown this can result in a reduced sample size [27, 28]. However, providing these assessments are consistently performed on a standardized day, either dialysis or non-dialysis day, and comparisons across providers are adjusted for demography, the timing of the assessment in relation to dialysis day of the week does not need to be standardised.

## Supporting information

**S1 Table. Symptom severity stratified by HD day (dialysis day of the week: HD1, HD2, HD3).**
(DOCX)

**S2 Table. Symptom score determined by baseline characteristics adjusted for dialysis day of the week (HD1, HD2, HD3).**
(DOCX)

**S3 Table. Baseline patient characteristics of participants who completed 1 instrument and >1 instruments throughout the study.**
(DOCX)

**S4 Table. 5 common symptoms comparing their severity at baseline for participants who completed only 1 instrument and >1 instruments throughout the study.**
(DOCX)

**S5 Table. Multivariable mixed effects linear regression comparing each hemodialysis day after long break, each HD day from HD1, HD day vs non-HD day.**
(DOCX)

**S6 Table. STROBE check list for observational study.**
(DOCX)

## Acknowledgments

The study team wish to acknowledge and thank the following contributing team members: Site principle investigators: Veena Reddy: Sheffield Teaching Hospital NHS Foundation Trust; Sandip Mitra: Central Manchester Healthcare Trust; Saeed Ahmed: City Hospitals Sunderland NHS Foundation Trust; Paul Warwicker: East & North Hertfordshire NHS Trust; Nicola Kumar: Guy's & St Thomas NHS Foundation Trust; Joyti Baharani: Heart of England Foundation Trust; Elizabeth Garthwaite: Leeds teaching Hospitals NHS Trust, Babu Ramakrishna: The Royal Wolverhampton NHS Trust, Albert Power: North Bristol NHS Trust; Mark Lambie: University Hospital of North Midlands NHS Trust; Alastair Ferraro: Nottingham University Hospitals NHS Trust; Implementation and research team members: Joanna Blackburn (qualitative research): Barnsley Hospital NHS Foundation Trust; Paul Harriman (quality improvement), Megan Bennett and Richard Simmonds (administrative support); Catherine Stannard & George Swinnerton (Think Kidneys) for processing the Your Health Survey; Sheffield Teaching Hospitals NHS Foundation Trust (Sponsor); Strategic advice from Michael Nation: Kidney Research UK. Prof Sue Mawson for chairing the evaluation advisory board. NIHR CRN research nurses at participating sites for consenting patients and supporting questionnaire completion.

## Author Contributions

**Conceptualization:** Pann Ei Hnynn Si, Steven Ariss, Stephen J. Walters, Martin Wilkie, James Fotheringham.

**Data curation:** Pann Ei Hnynn Si, Tania Barnes, Louese Dunn, Sonia Lee, James Fotheringham.

**Formal analysis:** Pann Ei Hnynn Si.

**Investigation:** Louese Dunn.

**Methodology:** Pann Ei Hnynn Si, Rachel Gair, Steven Ariss, Stephen J. Walters, Martin Wilkie, James Fotheringham.

**Project administration:** Tania Barnes, Louese Dunn, Sonia Lee.

**Resources:** Pann Ei Hnynn Si, Rachel Gair, Louese Dunn, Sonia Lee.

**Software:** Pann Ei Hnynn Si.

**Supervision:** Martin Wilkie, James Fotheringham.

**Validation:** Pann Ei Hnynn Si.

**Visualization:** Pann Ei Hnynn Si.

**Writing – original draft:** Pann Ei Hnynn Si.

**Writing – review & editing:** Pann Ei Hnynn Si, Rachel Gair, Tania Barnes, Louese Dunn, Sonia Lee, Steven Ariss, Stephen J. Walters, Martin Wilkie, James Fotheringham.

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
