## [Decision Letter · Decision Letter 0]

7 Feb 2022

PONE-D-21-25372Symptom burden according to dialysis day of the week in three times a week Haemodialysis patientsPLOS ONE

Dear Dr. Hnynn Si,

Thank you for submitting your manuscript to PLOS ONE. After careful consideration, we feel that it has merit but does not meet PLOS ONE’s publication criteria as it currently stands.  Please be aware that both referees (see their comments below) raised several important issues that must be thoroughly and unequivocally addressed before further consideration can possibly be given. Please submit your revised manuscript by Mar 24 2022 11:59PM. If you will need more time than this to complete your revisions, please reply to this message or contact the journal office at plosone@plos.org. Please include the following items when submitting your revised manuscript:A rebuttal letter that responds to each point raised by the academic editor and reviewer(s). You should upload this letter as a separate file labeled 'Response to Reviewers'.A marked-up copy of your manuscript that highlights changes made to the original version. You should upload this as a separate file labeled 'Revised Manuscript with Track Changes'.An unmarked version of your revised paper without tracked changes. You should upload this as a separate file labeled 'Manuscript'.

We look forward to receiving your revised manuscript.

Kind regards,

Gianpaolo Reboldi, MD, MSc, PhD

Academic Editor

PLOS ONE

Journal Requirements:

2. Please include the names of each of the renal centres in the study

“The Health Foundation (Scaling Up Round 2) funded the SHAREHD study and had no role in this study design, data collection, analysis, interpretation, or writing of the report.”

“PH conducts research funded by Vifor Pharma. JF has received speaker honoraria from Fresenius medical care, and conducts research funded by the National Institute of Health Research (NIHR), Vifor Pharma and Novartis. MEW has received speaker honoraria Fresenius and Baxter, has acted on an advisory board for Baxter and has conducted research funded by NIHR. SJW has received book royalties from Wiley and has received funds from NIHR, the Department of Health and Medical Research Council.”

5. We note that you have stated that you will provide repository information for your data at acceptance. Should your manuscript be accepted for publication, we will hold it until you provide the relevant accession numbers or DOIs necessary to access your data. If you wish to make changes to your Data Availability statement, please describe these changes in your cover letter and we will update your Data Availability statement to reflect the information you provide

Reviewers' comments:

Reviewer's Responses to Questions

**Comments to the Author**

1. Is the manuscript technically sound, and do the data support the conclusions?

Reviewer #1: No

Reviewer #2: Partly

2. Has the statistical analysis been performed appropriately and rigorously? 

Reviewer #1: No

Reviewer #2: No

3. Have the authors made all data underlying the findings in their manuscript fully available?

Reviewer #1: No

Reviewer #2: No

4. Is the manuscript presented in an intelligible fashion and written in standard English?

Reviewer #1: Yes

Reviewer #2: Yes

5. Review Comments to the Author

Reviewer #1: The manuscript presents secondary analysis of longitudinal data obtained from the SHAREHD stepped wedge CRT. While the study objectives sound interesting, I have the following questions.

1. It is not clear where the actual study trial was published (please mention a Reference!). If not, this manuscript needs a thorough rewrite, describing the trial (using detailed CONSORT items), and the analysis conducted in that context.

2. Please state the desired sample size/power under which the original CRT was powered, and data generated, to have some context. Better to write a full paragraph on what sample size/power was considered, in light of the primary response variable.

3. Mixed-effects linear regression was used to model severity, considering it as a "continuous" variable, whereas, in reality, it should have been an ordinal (response) variable. Ordinal regresson under a generalized linear mixed modeling framework can be readily implemented in SAS, or R. Hence, the analysis require a through redo.

4. Was there no need of considering any interaction terms in the list of covariables controlled for? In particular, understanding the changing patterns with time, given the longitudinal nature of the study.

5. Statements such as effect size 0.1, p 0.014, 95% CI ....require proper punctuations throughout the manuscript. Please edit thoroughly; it doesn't read well in the current form.

Reviewer #2: GENERAL COMMENTS

- The way the study has been designed, it should be considered an observational cohort study – therefore ensure your manuscript adheres to STROBE reporting guidelines

- Please use the current international nomenclature for kidney disease published in the journal Kidney International (Levey et al 2020), such as kidney failure (https://www.kidney-international.org/article/S0085-2538(20)30233-7/fulltext)

- Avoid use of uncommon abbreviations (e.g. DoW)

- Write numbers 0-9 in full (e.g. line 119 ‘6 months’)

ABSTRACT

- Background: further clarify that the rationale for the study relates to the need to know to what extent time and place of PROM completion should be standardised for research and clinical assessment.

- Methods: explain what the trial aimed to investigate because it provides relevant background context to your data collection

- Methods: Readers may not know what POS-S Renal refers to – either write in full or use ‘symptom burden score’

- Results: provide some reference to facilitate interpretation of the reported EQ-5D utility score

- Conclusions: significance of ‘location of completion’ not clear from the findings reported in the Results section

INTRODUCTION

- Explain what SONG-HD is (line 100)

- The statement ‘There is expanding literature that PROMS are not only effected by psychosocial issues, stress, emotions, and co-morbidities, the environment in which the instrument is completed may influence the result’ (lines 104-106) forms a key part of the rationale for this study – it needs more convincing support, rather than a single reference to an article that’s nearly 20 years old.

- ‘Failure to account for any underlying differences in symptom severity due to the haemodialysis schedule or location of completion at data collection,symptom analysis could underestimate the impact of interventions in this patient group’ (lines 108-9): impact could either be under- or overestimated, so better to state that this ‘failure to account….’ may distort evaluations of effectiveness of interventions

- No mention of HRQoL in the study’s aim as formulated in lines 111-3.

METHODS

- ‘Patients participating in SHAREHD were asked to complete instruments at baseline, 6, 12, and 18 months at either at dialysis unit or at home’ (lines 134-5): provide more detail on the extent to which the time and place of PROMs completion was protocolised. If this aspect was not protocolised –i.e. patients were free to complete it when and where they chose—you need to acknowledge the potential impact of this on the relationship between dialysis day and PROMs scores in the Discussion. This is relevant because the time/place participants chose for completion may be related to your outcome measures of interest: people’s likelihood of completing a PROM on a dialysis or non-dialysis day may depend on their symptom burden or QoL at that time (e.g. if their symptoms are worse on non-dialysis days, they may choose not to complete a PROM on those days because they are too poorly and wait for a dialysis day when they tend to feel better).

- ‘We excluded patients who had missing data’ (line 132) – were people excluded if they had any missing data or only when they had a certain level of missingness? Please provide more detail.

- Related to the previous point: provide information on how characteristics of people compared between those in- and excluded in the analysis. Also, if it appears you excluded a substantial number of participants because of missing outcome/PROMs measurements, please reflect in the Discussion on the fact that if and when people complete a PROM may be associated to their symptom burden/QoL, which in turn would impact on your findings (also see my first comment under METHODS).

- In the conclusion section of the abstract, the ‘location of completion’ seems central to the study’s key findings. Yet, it is unclear from the Methods how this aspect was defined, recorded and analysed, and there are no findings presented in the Results that are clearly and explicitly linked to and supporting this conclusion.

- Statistical analysis: State what level at which you considered a finding statistically significant. This level should be corrected for multiple testing (e.g. using Bonferroni correction), especially considering the large number of tests presented in Tables 3 and 4.

- Statistical analysis: related to the previous point, please try to reduce the number of tests, in particular those presented in Table 4. For example, you could consider running an ANOVA to explore whether there was any difference between HD1-3 for each symptom, and only compare specific days if the ANOVA test suggested this was the case. This would enable you to reduce the number of tests for most symptoms in Table 4 from 4 to 2.

- Statistical analysis: the intervention evaluated in the SHAREHD trial was likely to have had an impact on PROMs scores. Therefore, the time in the trial when PROMs measurement were taken (i.e. during control, intervention or sustainability period) needs to be accounted for in the analysis.

RESULTS

- Figure 1: it seems that no participants were excluded due to missing data – is this correct? Please clarify in the text and in the figure.

- Provide footnotes to Table 1 to explain all abbreviations in the table, as well as what the higher education levels refer to, and to clarify that higher Charlson scores represent more comorbidities

- I’m unclear what the relevance is of the analysis presented in Table 3 in relation to the study’s aim. The way it’s been presented seems to suggest the study is interested in exploring the relationship between patient characteristics and symptom scores, rather than between time/place of PROMs completion and PROMs scores – please provide a rationale for this analysis in the Methods and more explicitly relate the findings presented in the Results to the study aim.

DISCUSSION

- ‘Like other studies, we demonstrate that symptom severity is also susceptible tochange, in this study over a period of 18 months’ (line 297-8). This was what you presumably were trying to demonstrate in your SWT but not in the current study. Please clarify this.

- ‘We assume that completion of instruments on non-dialysis days occurred at home, and are unable to determine the timing of the completion of the instruments in relation to the dialysis session. However, it is our experience that most patients completed the instruments once established on the dialysis machine’ (lines 312-5). Clarify this assumption earlier on (in Methods and Results) and reflect in the Discussion how uncertainty within this assumption may have affected your findings and conclusions.

- Related to the previous point: specify what you mean with ‘timing of the completion of the instruments in relation to the dialysis session’ – it seems to refer to the exact timing of completion on a HD day. If this is this case, please clarify why this is relevant to mention as a limitation.

- ‘The high proportion of patients who change the severity of their response…’ (lines 318-9). Apart from it being better to talk about ‘report more severe symptoms’ instead of ‘change the severity of their response’, I’m unsure of the point you’re trying to make. Please consider to rephrase and more clearly link it to the study’s aim and the findings you presented.

- Please include some reflections on how your study and findings relate to the ‘expanding literature’ mentioned in the Introduction (lines 104-6)

6. PLOS authors have the option to publish the peer review history of their article (what does this mean?). If published, this will include your full peer review and any attached files.

Reviewer #1: No

Reviewer #2: **Yes: **Sabine N van der Veer

---

## [Author Response · Author response to Decision Letter 0]

6 May 2022

I've submitted the separate word document for responses to the reviewers. Some of the tables aren't included in the responses below due to formatting issues but those said tables were included in the word document attached with this submission. 

Comments from Editor

We have ensured the revised manuscript adhered to PLOS One’s journal requirement and used PACE digital diagnostic tool for the figures included in this revised manuscript. 

2. Please include the names of each of the renal centres in the study

We have included the names of 12 renal centres in the method section and study design.

“The Health Foundation (Scaling Up Round 2) funded the SHAREHD study and had no role in this study design, data collection, analysis, interpretation, or writing of the report.”

We have stated this disclosure in the manuscript as well as in cover letter. "The funders had no role in study design, data collection and analysis, decision to publish, or preparation of the manuscript."

“PH conducts research funded by Vifor Pharma. JF has received speaker honoraria from Fresenius medical care, and conducts research funded by the National Institute of Health Research (NIHR), Vifor Pharma and Novartis. MEW has received speaker honoraria Fresenius and Baxter, has acted on an advisory board for Baxter and has conducted research funded by NIHR. SJW has received book royalties from Wiley and has received funds from NIHR, the Department of Health and Medical Research Council.”

We have added this statement to the cover letter. 

Please see the response to question number 6. 

A minimal dataset required to reach the conclusions drawn from this manuscript required the linkage of identifiable patient information collected during the trial to Hospital Episode Statistics data, which at the time of writing is provided by the NHS Digital Data Access Request Service (NHS DARS, https://digital.nhs.uk/services/data-access-request-service-dars), and then appropriate processing. An application to NHS DARS can be submitted detailing lawful processing of the combined dataset and the period which HES data is required for. NHS DARS would verify appropriate permissions were in place as a result of this process. A data sharing agreement between the relevant parties would allow data to be transferred from the University of Sheffield to NHS DARS and on to those wishing to perform the enclosed analyses. Please contact ctru@sheffield.ac.uk for further information about the unlinked dataset which has the personal information required for linkage. 

In response to Reviewer 1: 

1. It is not clear where the actual study trial was published (please mention a Reference!). If not, this manuscript needs a thorough rewrite, describing the trial (using detailed CONSORT items), and the analysis conducted in that context.

It was published during the submission period of this manuscript to PLOS ONE. We have updated the references to this trial publication: A breakthrough series collaborative to increase patient participation with hemodialysis tasks: A stepped wedge cluster randomised controlled trial (plos.org) Reference number 14 in Manuscript.

2. Please state the desired sample size/power under which the original CRT was powered, and data generated, to have some context. Better to write a full paragraph on what sample size/power was considered, in light of the primary response variable.

We have included an abridged sample size/power calculation from original CRT (Line number 143-149). A full link to the trial was added in a reference. (Reference number 14) 

“The informing clinical trial sample size to determined using a recommended ICC value of 0.05 (20), to have a 90% power to detect an increase in the proportion undertaking five or more haemodialysis-related tasks from 15% to 30% as statistically significant at the 5% two-sided level: 12 clusters of 25 patients, with six clusters randomised at each step of SWCRT was arrived at. In recognition of the background mortality and renal transplantation rate and to mitigate the risk of incomplete data collection, the target recruitment per participating site was doubled to 50 (14).”

3. Mixed-effects linear regression was used to model severity, considering it as a "continuous" variable, whereas, in reality, it should have been an ordinal (response) variable. Ordinal regression under a generalized linear mixed modelling framework can be readily implemented in SAS, or R. Hence, the analysis require a through redo.

Thank you for highlighting this, which we acknowledge. This analysis was informed by the skills, software and time available within the research group and used as a vehicle to train the first author. Prior to the use of mixed effects linear regression, we recognised that an ordered mixed effects logistic regression could be applied, and trialled this in STATA. This was primarily to consider if linearity could be assumed, and reviewed the cut-points of the underlying latent variable (in the analysis presented below, pain). These largely incremented by constant values leading us to believe linearity could be assumed. 

Integration method: mvaghermite Integration pts. = 7

 Wald chi2(1) = 2.06

Log likelihood = -1608.2202 Prob > chi2 = 0.1517

 YSQ1 | Odds ratio Std. err. z P>|z| [95% conf. interval]

-------------+----------------------------------------------------------------

 HD Day | .9387223 .0414065 -1.43 0.152 .8609761 1.023489

-------------+----------------------------------------------------------------

 /cut1 | -1.09009 .1953785 -1.473024 -.7071547

 /cut2 | .3218006 .1913726 -.0532829 .6968841

 /cut3 | 1.980382 .2056775 1.577262 2.383503

 /cut4 | 4.218393 .2698197 3.689556 4.74723

-------------+----------------------------------------------------------------

SHAREHD_ID |

 var(_cons)| 2.756961 .4098724 2.06008 3.689583

Note: Estimates are transformed only in the first equation to odds ratios.

LR test vs. ologit model: chibar2(01) = 174.86 Prob >= chibar2 = 0.0000

In addition, the output does not lend itself to an assessment of what is clinically meaningful for the patient or clinical audience for the following reasons:

a) The odds ratio of moving from one level to another does not allow the reader to readily link this to the scale of the PROM

b) The estimate (odds ratio) does not allow linkage to a measure of clinically meaningful difference (either a whole level movement or an estimate based on the half-SD rule).

Finally, the second reviewer did propose other linear methods to screen for differences.

4. Was there no need of considering any interaction terms in the list of covariables controlled for? In particular, understanding the changing patterns with time, given the longitudinal nature of the study.

Thank you – this is an important topic for the renal community. To acknowledge the original nature of the SWCRT and therefore factor time, we have re-run the model adjusted for the steps (control and intervention). Symptom score did not change statistically significantly comparing the control period and intervention period. Analysis in relation to dialysis day of the week adjusted for control and intervention period did not result in change in symptom score. By including steps, effectively we also have between 6-12 months differentiation in time. 

Because of the number of symptoms and attendant models informing this manuscript we did not feel it was appropriate to try and incorporate any inference around how symptoms change over time – this is a large topic and we hope to report something specific around this in the future. We were concerned that adding interactions could potentially add more type 1 errors.

5. Statements such as effect size 0.1, p=0.014, 95% CI -0.02 to 0.2 etc. ....require proper punctuations throughout the manuscript. Please edit thoroughly; it doesn't read well in the current form.

We have edited the results in the format that was consistent throughout manuscript.  

Reviewer 2: 

GENERAL COMMENTS

1. The way the study has been designed; it should be considered an observational cohort study – therefore ensure your manuscript adheres to STROBE reporting guidelines

We have submitted a STROBE reporting guideline checklist in supplementary material (S4 table)

2. Please use the current international nomenclature for kidney disease published in the journal Kidney International (Levey et al 2020), such as kidney failure (https://www.kidney-international.org/article/S0085-2538(20)30233-7/fulltext)

 Avoid use of uncommon abbreviations (e.g. DoW) 

 Write numbers 0-9 in full (e.g. line 119 ‘6 months’)

We have edited and changed the abbreviations and numbers as reviewer 2 suggested. 

ABSTRACT

3. Background: further clarify that the rationale for the study relates to the need to know to what extent time and place of PROM completion should be standardised for research and clinical assessment.

Due to word limitation for abstract, some of this detail was previously removed but we have edited to clarify the aim of this study with the following: Line number 30-32

“The relationship between dialytic interval and patient reported outcome measures (PROM) has not been quantified and the extent to which dialysis day of PROM completion need to be standardised for is unknown.” 

4. Methods: explain what the trial aimed to investigate because it provides relevant background context to your data collection

We have added some background info of SHAREHD in the abstract. 

“Three times a week haemodialysis patients participating in a stepped wedge trial to increase patient participation in haemodialysis tasks completed PROMs (POS-S Renal symptom score and EQ5D-5L) at recruitment, six, 12 and 18 months.” Line number 33-35

5. Methods: Readers may not know what POS-S Renal refers to – either write in full or use ‘symptom burden score’

We have addressed these instruments as PROMS symptom score in the abstract. 

“PROMs (POS-S Renal symptom score and EQ5D-5L)” Line number 34-35

6. Results: provide some reference to facilitate interpretation of the reported EQ-5D utility score

We have removed the EQ5D utility score from abstract due to the 300 word limitation and result section focused mainly on the main topic of HD schedule and symptoms but it remained in main manuscript. 

7. Conclusions: significance of ‘location of completion’ not clear from the findings reported in the Results section

Location of completion is not the same as non-dialysis day of the week (this is an assumption). 70% of patients assigned to non- HD group completed questionnaires at home. We have changed the language to non-dialysis day instead of location. 

INTRODUCTION

8. Explain what SONG-HD is (line 100)

I’ve expanded SONG_HD and its purpose. 

“When Standardized Outcome in Nephrology (SONG-HD) , consensus group to establish core outcome to be measured and reported in haemodialysis trial,” Line number 74-75

9. The statement ‘There is expanding literature that PROMS are not only affected by psychosocial issues, stress, emotions, and co-morbidities, the environment in which the instrument is completed may influence the result’ (lines 104-106) forms a key part of the rationale for this study – it needs more convincing support, rather than a single reference to an article that’s nearly 20 years old.

There are more contemporary evidence-based literature on effects of psychosocial issues, stress, emotions, patients characteristics influencing PROMS measures which we have referenced. We were only able to identify the very old reference of the effect of environment. We have moved away from the issue of environment to make the focus of the manuscript the schedule.

10. ‘Failure to account for any underlying differences in symptom severity due to the haemodialysis schedule or location of completion at data collection, symptom analysis could underestimate the impact of interventions in this patient group’ (lines 108-9): impact could either be under- or overestimated, so better to state that this ‘failure to account….’ may distort evaluations of effectiveness of intervention

We completely agree that bias can be both ways either over or underestimated and amended this statement in revised manuscript. 

“Failure to account for any underlying differences in symptom severity due to the day of instrument completion in relation to the dialysis schedule could bias the impact and effectiveness of interventions in this patient group”. Line number 84-86

11. No mention of HRQoL in the study’s aim as formulated in lines 111-113. 

Although we have led the introduction with symptom burden and effect on HRQoL in HD patients, this study aim is to explore the effect of symptom burden as a mean of HRQoL in relation to HD day of the week. 

“To quantify this, we used data from a large stepped wedge randomised controlled trial with aim to determine the association between symptom burden and the haemodialysis schedule in three times a week haemodialysis patients and to explore the effect of PROMS completion in dialysis and non dialysis days, accounting for patient characteristics.” Line number 86-89

Methods

12. ‘Patients participating in SHAREHD were asked to complete instruments at baseline, 6, 12, and 18 months at either at dialysis unit or at home’ (lines 134-5): provide more detail on the extent to which the time and place of PROMs completion was protocolised. If this aspect was not protocolised –i.e. patients were free to complete it when and where they chose—you need to acknowledge the potential impact of this on the relationship between dialysis day and PROMs scores in the Discussion. This is relevant because the time/place participants chose for completion may be related to your outcome measures of interest: people’s likelihood of completing a PROM on a dialysis or non-dialysis day may depend on their symptom burden or QoL at that time (e.g. if their symptoms are worse on non-dialysis days, they may choose not to complete a PROM on those days because they are too poorly and wait for a dialysis day when they tend to feel better).

Research protocol did not specify where to compete the questionnaires. But participants were asked to tick where they completed the instruments: either at home, dialysis unit or clinic. It is possible that frailer people might take instruments home to complete. But our analyses showed that only 16.7% of older people (>65yr) completed the instruments on non-dialysis days and 12.8% of patients with high CCI score (score >5) completed on non-dialysis days. In addition, our recommendation mitigates against this (older patients >65 have less symptoms in our analysis). I've included the relevant analyses and tables in the word documents attached with this submission (document labeled as responses to reviewers)

13. ‘We excluded patients who had missing data’ (line 132) – were people excluded if they had any missing data or only when they had a certain level of missingness? Please provide more detail.

We excluded missing data for the adjustment covariates and mechanism of missingness for comorbidity is failure to link via NHS Digital – assumed at random.

14. Related to the previous point: provide information on how characteristics of people compared between those in- and excluded in the analysis. Also, if it appears you excluded a substantial number of participants because of missing outcome/PROMs measurements, please reflect in the Discussion on the fact that if and when people complete a PROM may be associated to their symptom burden/QoL, which in turn would impact on your findings (also see my first comment under METHODS).

We have 62 participants who completed only 1 instrument throughout the study period. Comparing this cohort with patients who completed the instruments more than once in follow up period, patient characteristics are similar (perhaps slightly younger and less co morbid) and severity of symptom burden at baseline is less (we have provided 5 most prevalence symptoms as an example). Therefore, we did not think this will impact the outcome of the analysis we performed to demonstrate the relationship between symptom burden and dialysis day of the week. I've included the relevant analyses and tables in the word documents attached with this submission (document labeled as responses to reviewers)

15. In the conclusion section of the abstract, the ‘location of completion’ seems central to the study’s key findings. Yet, it is unclear from the Methods how this aspect was defined, recorded and analysed, and there are no findings presented in the Results that are clearly and explicitly linked to and supporting this conclusion.

Majority of patients who were assigned to non HD days completed instruments at home and HD days at renal units. We changed the language to non HD day in line with wider question of dialysis schedule. 

16. Statistical analysis: State what level at which you considered a finding statistically significant. This level should be corrected for multiple testing (e.g. using Bonferroni correction), especially considering the large number of tests presented in Tables 3 and 4.

We performed multiple comparison testing with different variable and therefore, we agreed that P value should be lowered. But given that most results were negative, assumption of P value might probably caveat. Based on our analysis, we don’t need to adjust for Dialysis day of the week. Therefore, we believe a Bonferroni correction would not change that and if anything, its significant will be less significant. 

17. Statistical analysis: related to the previous point, please try to reduce the number of tests, in particular those presented in Table 4. For example, you could consider running an ANOVA to explore whether there was any difference between HD1-3 for each symptom, and only compare specific days if the ANOVA test suggested this was the case. This would enable you to reduce the number of tests for most symptoms in Table 4 from 4 to 2.

We agree that there are a lot of tests and that this introduces a range of problems. Although an ANOVA can be performed to compare symptoms change between HD1 vs HD2/HD3 (e.g. categorical vs continuous), it does not fully capitalise on the statistical efficiency of treating HD days as continuous variable as we do in one subset of our models. 

Although doing many tests does introduce the risk of a type I error, we have generally said that the timing of assessment in relation to the dialysis schedule does not need to be accounted for (which is more likely to be a type II error)

The other threat of so many models is reader fatigue. In order to reduce the number of tables, we have reported these results in figure form in the main manuscript and I’ve submitted the table as supplementary material. Figure 3 and S3 Table

18. Statistical analysis: the intervention evaluated in the SHAREHD trial was likely to have had an impact on PROMs scores. Therefore, the time in the trial when PROMs measurement were taken (i.e. during control, intervention or sustainability period) needs to be accounted for in the analysis.

We have re-run the model adjusted for steps (control and intervention). Symptom score did not change statistically significantly comparing the control period and intervention period and these are described in the manuscript (Line number 248-250) . Analysis in relation to dialysis day of the week adjusted for control and intervention period did not result in change in symptom score. 

“Symptom score comparing the control period and intervention period predicting the symptom severity independently in this model did not reach statistical significance”.

RESULTS

19. Figure 1: it seems that no participants were excluded due to missing data – is this correct? Please clarify in the text and in the figure.

We excluded missing data for the adjustment covariates and but no missing data for PROMS instrument itself. We have edited the flow diagram figure for more clarification. 

20. Provide footnotes to Table 1 to explain all abbreviations in the table, as well as what the higher education levels refer to, and to clarify that higher Charlson scores represent more comorbidities

We have expanded the abbreviations in table and foot notes added to clarify education levels and comorbidities. 

21. I’m unclear what the relevance is of the analysis presented in Table 3 in relation to the study’s aim. The way it’s been presented seems to suggest the study is interested in exploring the relationship between patient characteristics and symptom scores, rather than between time/place of PROMs completion and PROMs scores – please provide a rationale for this analysis in the Methods and more explicitly relate the findings presented in the Results to the study aim.

We included this relationship between patient characteristics and symptoms so that others designing the clinical trials can refer to it in order to assist with power calculations. This will mandate their adjustment in observational studies and potentially in clinical trials as studies have shown this can result in a reduced sample size. We appreciate the reviewer’s point and have moved this table to supplementary material to reduce the number of tables as we agreed that these finding are not directly linked to the study aim. 

DISCUSSION

22. ‘Like other studies, we demonstrate that symptom severity is also susceptible to change, in this study over a period of 18 months’ (line 297-8). This was what you presumably were trying to demonstrate in your SWT but not in the current study. Please clarify this.

We agreed that the main purpose of this study is not to demonstrate the change in symptom burden over 18 months. We demonstrated symptoms change over a week according to dialysis schedules although it was not statistically significant. We apologise that this statement introduces confusion. Therefore, we decided not to include this statement in our revised manuscript. 

23. ‘We assume that completion of instruments on non-dialysis days occurred at home, and are unable to determine the timing of the completion of the instruments in relation to the dialysis session. However, it is our experience that most patients completed the instruments once established on the dialysis machine’ (lines 312-5). Clarify this assumption earlier on (in Methods and Results) and reflect in the Discussion how uncertainty within this assumption may have affected your findings and conclusions.

We have changed the language of location of completion to non-dialysis days and reflected this in discussion. Line number 308-311

“As the participants were allowed to complete the instruments on the day of their choice either on non-dialysis day or dialysis day and they may choose to complete these questionnaires on the day they feel better, this has potential impact on the outcome of symptom burden in relation to dialysis days.” 

24. Related to the previous point: specify what you mean with ‘timing of the completion of the instruments in relation to the dialysis session’ – it seems to refer to the exact timing of completion on a HD day. If this is this case, please clarify why this is relevant to mention as a limitation.

We have expanded this in discussion session. (line number 303-308)

“For instruments completed on a dialysis day we are unable to determine the timing of the completion of the instruments in relation to the dialysis session which can be either before or during or after dialysis session. This may be relevant as patients may feel more unwell immediately after dialysis due to dialysis process itself affecting their symptom burden. However, it is our experience that most patients completed the instruments once established on the dialysis machine.”

25. ‘The high proportion of patients who change the severity of their response…’ (lines 318-9). Apart from it being better to talk about ‘report more severe symptoms’ instead of ‘change the severity of their response’, I’m unsure of the point you’re trying to make. Please consider to rephrase and more clearly link it to the study’s aim and the findings you presented.

As we answered in question number 22, we agreed that this study aim was not to demonstrate change in symptom burden, therefore, we decided not to include this statement. 

26. Please include some reflections on how your study and findings relate to the ‘expanding literature’ mentioned in the Introduction (lines 104-6)

We added additional reflection and conclusion related to this statement. 

“Symptoms prevalence in this study is comparable to other studies (22) including a systematic review on symptoms prevalence in ESRF patients (23), both in prevalence, type of symptoms reported. How severity of symptom varies in relation to patient characteristics and comorbidities was also comparable (13) . Line number 285-288

“We identified significant associations between symptom severity in different age groups, comorbidity and time on dialysis, mandating their adjustment in observational studies and potentially in clinical trials as studies have shown this can result in a reduced sample size(26)(27).” Line number 312-315

---

## [Decision Letter · Decision Letter 1]

30 Jun 2022

PONE-D-21-25372R1Symptom burden according to dialysis day of the week in three times a week Haemodialysis patientsPLOS ONE

Dear Dr. Hnynn Si,

Thank you for submitting your manuscript to PLOS ONE. After careful consideration, we feel that it has merit but there are a few comments  raised during the review process that deserve consideration. Therefore, we invite you to submit a revised version of the manuscript that addresses the remaining minor points by referee #2.

We look forward to receiving your revised manuscript.

Kind regards,

Gianpaolo Reboldi, MD, MSc, PhD

Academic Editor

PLOS ONE

Journal Requirements:

Reviewers' comments:

Reviewer's Responses to Questions

**Comments to the Author**

1. If the authors have adequately addressed your comments raised in a previous round of review and you feel that this manuscript is now acceptable for publication, you may indicate that here to bypass the “Comments to the Author” section, enter your conflict of interest statement in the “Confidential to Editor” section, and submit your "Accept" recommendation.

Reviewer #1: All comments have been addressed

Reviewer #2: (No Response)

2. Is the manuscript technically sound, and do the data support the conclusions?

Reviewer #1: Yes

Reviewer #2: Yes

3. Has the statistical analysis been performed appropriately and rigorously? 

Reviewer #1: Yes

Reviewer #2: Yes

4. Have the authors made all data underlying the findings in their manuscript fully available?

Reviewer #1: Yes

Reviewer #2: (No Response)

5. Is the manuscript presented in an intelligible fashion and written in standard English?

Reviewer #1: Yes

Reviewer #2: No

6. Review Comments to the Author

Reviewer #1: (No Response)

Reviewer #2: Thank you for addressing my comments on your manuscript, which has now much improved. Some final, minor points:

* Your response to my point 11 ("No mention of HRQoL in the study’s aim") is unclear: what does 'effect of symptom burden as a mean of HRQoL' mean? I strongly recommend you say something about the role/importance of HRQoL at the end of the introduction, so that this doesn't come as a surprise to readers when they get to the Methods and Results (e.g. that it was a secondary outcome of interest?)

* Your response to my point 14 (“provide information on how characteristics of people compared between those in- and excluded in the analysis”) is adequate. However, I suggest you include the relevant analyses/tables as supplementary materials, so that interested readers also have access to this information.

* Your response to my point 16 (“State the level at which you considered a finding statistically significant”) is unclear: it’s hard to follow your argument for why statistical significance is (or isn’t?) relevant. But regardless of what you decide to use as a threshold, please clarify for readers in the manuscript (and not just for me in the response letter) what your definition of ‘statistical significance’ is and how you accounted for multiple testing (or why this is not required).

* There are several grammatical/style errors throughout the newly added sections in the abstract and main text - the manuscript therefore requires one, last thorough round of text editing by a native English speaker prior to submission.

7. PLOS authors have the option to publish the peer review history of their article (what does this mean?). If published, this will include your full peer review and any attached files.

Reviewer #1: No

Reviewer #2: **Yes: **Sabine N van der Veer

---

## [Author Response · Author response to Decision Letter 1]

13 Aug 2022

REBUTTAL LETTER FOR SUBMISSION OF REVISED MANUSCRIPT

To The Editor in Chief, 

PLOS ONE Journal Date: 11/08/2022

We are grateful for your and the reviewers’ comment on the manuscript “Symptom Burden According To Dialysis Day Of The Week In Three Times A Week Haemodialysis Patients “. We’ve revised and modified the manuscript according to reviewers’ critiques. In the following, we address each of their comment.

Journal Requirements:

I’ve reviewed the reference list and they are complete. There was no citation of retracted papers. 

Response to reviewer 2

1. Your response to my point 11 ("No mention of HRQoL in the study’s aim") is unclear: what does 'effect of symptom burden as a mean of HRQoL' mean? I strongly recommend you say something about the role/importance of HRQoL at the end of the introduction, so that this doesn't come as a surprise to readers when they get to the Methods and Results (e.g. that it was a secondary outcome of interest?)

We’ve added the role of HRQoL in this study. (line number 83-88)

“HRQoL has been shown to be a predictor of morbidity and mortality in haemodialysis patients (14)(15) and HRQoL measures play an important role in evaluating cost effectiveness of treatment. Failure to account for any underlying differences in symptom severity due to the day of instrument completion in relation to the dialysis schedule could bias the impact and effectiveness of interventions for symptoms and HRQoL, which have been prioritised by the patients and clinicians(2) and could lead to failure of new treatment or interventions to be approved.”

2. Your response to my point 14 (“provide information on how characteristics of people compared between those in- and excluded in the analysis”) is adequate. However, I suggest you include the relevant analyses/tables as supplementary materials, so that interested readers also have access to this information.

We’ve added this in result section and provided supplementary tables. (line number 234-237 and Supplementary table 3 and 4).

“There were 62 participants who completed only one instrument throughout the study period. Comparing this cohort with patients who completed the instruments more than once in follow up period, patient characteristics and severity of symptom burden at baseline were similar (S3 Table, S4 Table).”

3. Your response to my point 16 (“State the level at which you considered a finding statistically significant”) is unclear: it’s hard to follow your argument for why statistical significance is (or isn’t?) relevant. But regardless of what you decide to use as a threshold, please clarify for readers in the manuscript (and not just for me in the response letter) what your definition of ‘statistical significance’ is and how you accounted for multiple testing (or why this is not required).

We’ve clarified this in method as well as in discussion section. (line number 178, 309-313)

“P value < 0.05 was considered the threshold for statistical significance.”

“We performed multiple testing with different variables with statistical significance considered as a p value less than 0.05, and concluded adjustment for dialysis day of the week is not required as most of the results were non-significant. Correction of p values for multiple testing would further reduce significance, not altering our conclusions, and is therefore is not required.”

4. There are several grammatical/style errors throughout the newly added sections in the abstract and main text - the manuscript therefore requires one, last thorough round of text editing by a native English speaker prior to submission.

We hope we have identified and corrected the errors.

---

## [Editor Report · Decision Letter 2]

1 Sep 2022

Symptom burden according to dialysis day of the week in three times a week Haemodialysis patients

PONE-D-21-25372R2

Dear Dr. Hnynn Si,

We’re pleased to inform you that your manuscript has been judged scientifically suitable for publication and will be formally accepted for publication once it meets all outstanding technical requirements.

Kind regards,

Gianpaolo Reboldi, MD, MSc, PhD

Academic Editor

PLOS ONE
---

## [Editor Report · Acceptance letter]

6 Sep 2022

PONE-D-21-25372R2 

Symptom Burden According To Dialysis Day Of The Week In Three Times A Week Haemodialysis Patients 

Dear Dr. Hnynn Si:

I'm pleased to inform you that your manuscript has been deemed suitable for publication in PLOS ONE. Congratulations! Your manuscript is now with our production department. 

Kind regards, 

on behalf of

Prof Gianpaolo Reboldi 

Academic Editor

PLOS ONE